# Supporting children and young people when making decisions about joining clinical trials: qualitative study to inform multimedia website development

Jacqueline M Martin-Kerry,[1] Peter Knapp,[2] Karl Atkin,[1] Peter Bower,[3] Ian Watt,[2] Catherine Stones,[4] Steven Higgins,[5] Rebecca Sheridan,[1] Jenny Preston,[6,7] Danielle Horton Taylor,[8] Paul Baines,[9] Bridget Young[10]

For numbered affiliations see end of article.

**Correspondence to**
Dr Jacqueline M Martin-Kerry; jackie.martin-kerry@york.ac.uk

## ABSTRACT

**Objectives** To understand stakeholders' views regarding the content and design of paediatric clinical trial multimedia websites. To describe how this knowledge informed the development of the multimedia websites.

**Design** Qualitative study comprising two rounds of interviews or focus groups, with thematic analysis of interview transcripts.

**Participants** Sixty-two people (21 children and young people with long-term health conditions, 24 parents and 17 professionals).

**Setting** One UK children's hospital and one UK Young Persons' Advisory Group.

**Results** When asked what was important in deciding whether to join a trial, children, young people and parents prioritised information about what participation would involve, what the trial was testing, potential benefits and risks of participation and knowing they could leave the trial if they later changed their minds. Young people and parents trusted trial teams to follow regulatory and quality requirements and therefore did not think such information was a priority for the websites, although logos of trusted organisations could lend credibility. Professionals largely concurred with these views. Children and young people advised on the importance of designing the multimedia website to ensure its appearance, tone and wording suited the intended audience and on using animated characters to facilitate children's engagement.

**Conclusions** Our study provides insights into the information that families value when deciding about healthcare trial participation. It provides guidance on the design of information resources to appeal to children and young people, while also being acceptable to parents and professionals who are often gatekeepers of children's access to information. Our findings will be of use to others developing similar multimedia websites. We report specific information needs and new visual preferences that are not usually addressed in printed trial information. Our work illustrates what qualitative research and participatory design practices can contribute to the development of information resources more generally.

**Trial registration number** ISRCTN73136092; Pre-results.

## Strengths and limitations of this study

► This is the first UK study to explore and apply knowledge about stakeholders' views to inform the development of multimedia websites that aim to help children and young people decide about clinical trial entry.
► We worked to make the research process interactive and engaging, particularly for children and young people, by using activities and examples of websites, animations and characters.
► Limitations included that some of the younger children had difficulty engaging during the interviews.
► Sampling was limited to two regions of the UK, and we did not interview many children and young people with long-term health conditions aged between 9 years and 14 years. However, we interviewed children, young people and parents with trial experience as well as those who had not previously participated in a trial.
► The findings enabled us to develop the multimedia websites to suit the needs and preferences of children, young people and their parents and identify ways that the multimedia websites could enhance their engagement in decisions about trials.

## INTRODUCTION

High-quality clinical trials involving children are essential to ensure that medication and treatments for children are effective and safe.[1–4] Conventionally, participant information about trials is provided in printed form. These documents should be understandable to potential participants and assist their decision making.[5] It is recognised that children and young people should have opportunities to understand what any research would entail and participate in the decision-making process.[6] However, as with adult studies,[7] printed participant information sheets for paediatric clinical trials are often lengthy and difficult to read and understand.[8 9] This

has implications for children and young people's understanding of trials, which may affect their decision to participate. Several studies have identified gaps in children's understanding about the purpose of trials and of what participation in a trial entails.[10 11] A recent review has highlighted the importance of providing information about research to children and young people directly, not just via parents, and advocates that printed information is written in a way that is 'appealing and understandable' to children and young people.[12] The limitations of printed media for informing potential trial participants about trials has been recognised by the UK Health Research Authority, who recommend exploration of the usefulness of other media.[13] A novel approach for providing information to potential trial participants is to use multimedia websites, which allow information to be presented using animation, video, text and images. Multimedia websites have been beneficial in informing potential participants and recruiting participants to studies,[11 14 15] although work describing the development of such resources for children and young people is limited.

This paper addresses this gap by reporting on a qualitative interview and focus group study with children, young people, parents and professionals to inform the development of the multimedia websites for use in paediatric clinical trials. This study was part of the TRials Engagement in Children and Adolescents (TRECA) project,[16] which also involves the development and subsequent evaluation of the impact of multimedia websites on recruitment, retention and decision-making in paediatric trials. The current qualitative study drew on the principles of participatory design to help ensure that the newly developed multimedia websites would meet the needs and preferences of children and young people being approached to participate in trials while also being acceptable to parents and professionals. The acceptability of such materials to parents and professionals is important as they are gatekeepers to children's access to information about clinical trials or treatment options.[17] Here we describe the findings from the qualitative study and how we used these to inform the development of the multimedia websites.

## METHODS

This study involved two rounds of interviews (individual, joint or focus group discussions). The first round focused on identifying participants' needs and preferences for information about clinical trials, while the second round sought their views of prototype multimedia websites that had been developed based on feedback from the first round of interviews. We aimed to involve participants in both rounds to facilitate the iterative development of the multimedia websites, with some replacement of participants for the second round, when required.

### Sampling and recruitment of participants

We recruited children, young people and parents through two routes: a children's hospital in North West

England and a Generation R Young Persons' Advisory Group (YPAG) located in the Midlands, comprising children and young people who advise on the design of research involving children and young people in the NHS. Recruitment via the hospital route involved nurses contacting families of children who were receiving specialised care for long-term conditions, either by phone or in person; the YPAG route involved the YPAG coordinator contacting members and their parents by text message or by phone call. The interviewer (JMM-K) telephoned those who agreed to contact to explain the study and arrange a time for an interview. Sampling of children, young people and parents aimed to encompass variation in age, gender, long-term health condition, trial experience and ethnicity. Professionals were initially emailed by an academic paediatrician working in the study hospital. Those who expressed an interest were subsequently contacted by JMM-K by email to arrange the interviews. Sampling of professionals aimed for diversity in roles and paediatric specialities within trial settings.

We provided participants with printed information sheets about the study, with different versions for children, young people, parents and professionals. Participants under 16 years provided assent, and their parents provided consent for the child or young person's participation. All participants received a £10 gift voucher after being interviewed.

### Data collection

We undertook topic-guided, semistructured interviews with children and young people with long-term health conditions, their parents and professionals. Interviews were conducted by the same interviewer (JMM-K) over the two rounds (July–October 2016 and November 2016–January 2017). JMM-K is a research lead on the study with a science background and training in qualitative research methodology; she had undertaken and analysed focus groups previously and BY and KA both provided advice and guidance. JMM-K explained at the start of interviews that all comments were welcomed and that a key aim of the interviews was to inform the development of the websites to suit potential end users. All interviews and focus groups were audio-recorded and transcribed; transcripts were checked and pseudoanonymised before analysis.

### First round of interviews

JMM-K began interviews narratively by asking participants about their experience of research and involvement in health research. Later in the interviews, she showed all participants 20 cards with topics that adults had previously identified as being important when deciding to take part in a trial[18] and asked them to rank the topics (see box 1) as 'important', 'somewhat important' and 'not important'.[19 20] JMM-K also asked participants to comment on their rankings and on any information not included in the cards that they felt was important when deciding whether to participate in a trial. Research is quite an abstract and challenging topic particularly

**Box 1  Topics within first round of interviews and focus group discussions**

► Why is the study happening?
► Why have I been invited?
► Do I have to take part?
► What will happen if I do not want to carry on with the study?
► What will happen to me if a take part?
► Will it cost me any money to take part?
► Will I be paid for taking part?
► What is being tested in the study (for example what drug or device)?
► What are the different treatments or types of healthcare being provided in the study?
► What are the risks of taking part?
► What are the possible benefits of taking part?
► What happens when the study ends?
► What would happen if a problem happens in the study/could I make a complaint?
► Will my taking part be kept confidential (private)?
► Will my general practitioner/family doctor be told about my taking part?
► What will happen to any blood tests or other samples that I have as part of the study?
► What will happen to the results of the study?
► Who is running the research?
► Who is paying for the research?
► Who has reviewed the study?

for younger participants, and this exercise helped to facilitate discussion and support participants in identifying what was important to them.[21] To similarly facilitate participants discussing their preferences regarding different designs of multimedia websites, JMM-K showed participants various examples of existing websites, animations and a video (see table 1 for details) and examples of character designs (figure 1) and asked them to comment on these. These examples represented a range of presentation styles and design styles.

Finally, we explored how the multimedia websites should look and function by asking participants to rank six cards, each with a printed statement related to criteria for assessing websites.[22] These covered content, structure and navigation, visual design, interactivity, functionality and credibility. An initial analysis of the first round of interviews informed the design of the prototype multimedia websites.

### Second round of interviews

In the second round of interviews, we explored participants' views of the prototype multimedia websites. We developed the topic guides based on input from a qualitative methodologist (BY) with experience of researching families' decisions about clinical trial participation and an education expert (SH). The websites concerned a paediatric diabetes trial,[23] with one version aimed at children aged 6–11 years and their parents and the other at young people aged 12 years upwards and their parents. The multimedia websites included two animations that were not age specific: a trial-specific 'explainer' animation that summarised the diabetes trial and a generic 'why do we do trials' animation that outlined the rationale for conducting a trial. We showed participants the animations and the multimedia website suited to their age (and the other version if they wished; see figures 2 and 3) and explored their views of these (see box 2 for prompts). We also invited suggestions to improve the multimedia websites and animations. We waited until this second round when we had concrete materials available before interviewing young children (6–8 years old) and their parents, as we felt the content covered in the first round of interviews would be too abstract to engage young children.

### Data analysis to inform the development of the multimedia websites

We conducted an initial rapid descriptive data analysis[24] in order to provide timely feedback to the company developing the multimedia websites. This analysis was deductive and focused on the content and design aspects of the website. Subsequent analyses were iterative and thematic with both inductive and deductive aspects.[25 26] This involved reading transcripts multiple times to aid

**Table 1  Resources used within the first round of interviews**

| Resource | Example of: | Link |
| --- | --- | --- |
| HeadSpace | Website | https://www.headspace.com/ |
| Toca | Website | https://tocaboca.com/ |
| Health Research: making the right decision for me (Nuffield Council on Bioethics) | Animation | https://www.youtube.com/watch?v=6yaKwLG_vlE |
| What is a randomised trial? (Cancer Research UK) | Animation | http://www.cancerresearchuk.org/about-cancer/find-a-clinical-trial/what-clinical-trials-are/randomised-trials |
| Can we tell which children with febrile neutropaenia have a bad bug or infection? (Dr Bob Phillips) | Animation (Lego) | https://www.youtube.com/watch?v=Z1AXzJqatds |
| Hi-Light Trial video | Video of child and parent talking about participating in the trial | Video not publicly available. Hi-Light protocol is available.[27] |

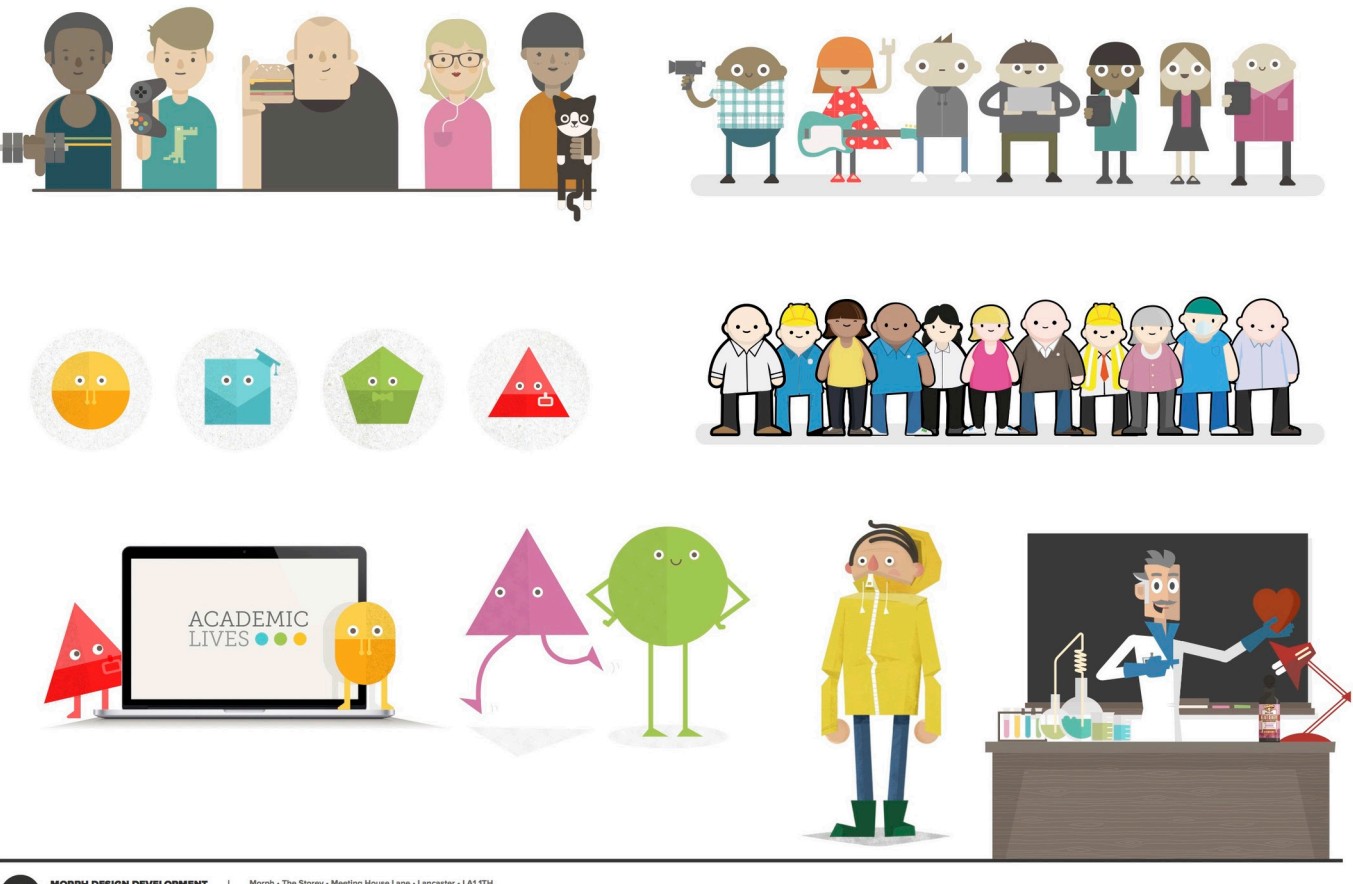

**Figure 1** Character design options.

familiarisation, followed by line-by-line coding to identify recurring ideas and organising these into themes and subthemes. We also compared interviews across participant groups to identify similarities and differences. Analysis was undertaken by JMM-K with guidance provided via regular meetings with BY and KA to discuss transcripts and aid data interpretation. As we outline below, findings from the data analysis were also discussed with the TRECA Study Patient and Parent Advisory Group to seek their thoughts on ways to implement feedback received from the qualitative study in the development of the multimedia websites. Data coding and indexing was assisted by Microsoft Excel 2010 software. JMM-K drafted several iterations of a data analysis report, which BY, KA and PK read and discussed.

### Patient involvement
Patient and public involvement informed the overall research questions within TRECA from its inception and particularly during the grant-writing stage. The TRECA Study also convened an active and engaged Patient and Parent Advisory Group to give input throughout the study and suggest ways to implement changes requested by participants in the qualitative study. One of the members (DHT) of the Patient and Parent Advisory Group is an author on this manuscript. Children, young people, parents and professionals who participated in this study

will be invited to attend a presentation about the findings of the study or to receive reader-friendly summaries of the study as they prefer.

## RESULTS
### Participants
Across the two rounds of interviews, a total of 87 people were invited to participate, and 62 were interviewed. Of the 87 people invited, 35 were professionals of whom 17 were interviewed; those who declined were either unavailable for interview or did not respond. Twenty-eight parents were invited with 24 taking part; and 24 young people were invited to participate with 21 participating. Those who declined did so either due to other commitments or because they did not want to participate. Of the 62 participants, 15 participated in round 1 only (6 children and young people, 6 parents and 3 professionals), 22 in round 2 only (10 children and young people, 10 parents and 2 professionals) and 25 in both rounds (5 children and young people, 8 parents and 12 professionals). Table 2 provides a summary of the participant characteristics and online supplementary appendix 1 shows detailed participant characteristics. Of the 21 children and young people interviewed, 16 preferred to be interviewed with a parent(s) present. The five young

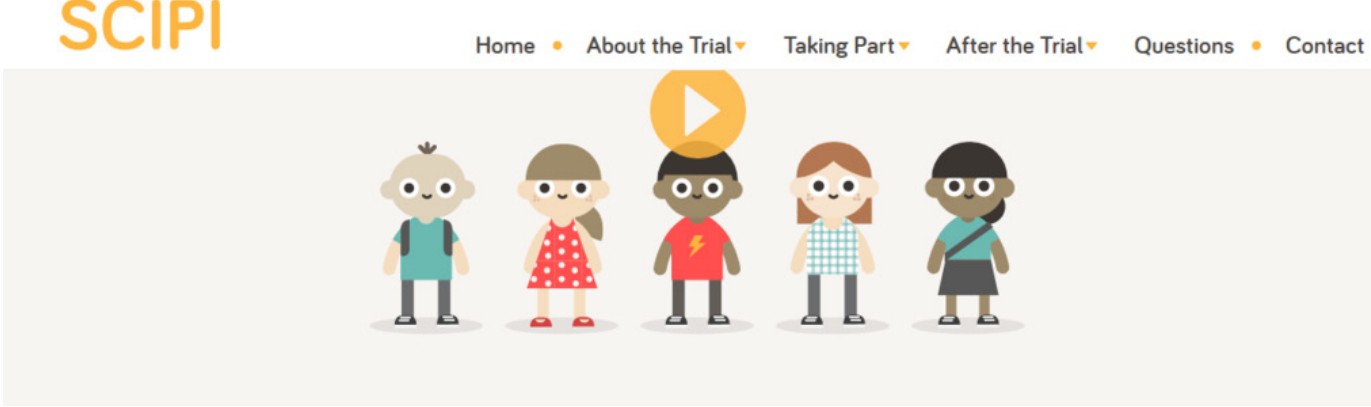

## About The Trial

The SCIPI trial will test two ways to give insulin to children and teenagers with diabetes. The two ways are an injection or a pump. Insulin helps you use sugar properly.

We hope this website will help you decide if you want to take part in the SCIPI trial.

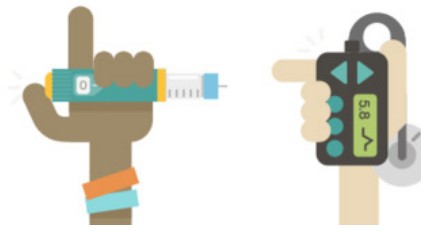

**Figure 2**  Website for children and their parents.

people who were members of the YPAG preferred to take part in focus group interviews, as did all but three professionals. Focus groups had between four and nine participants. One participant was interviewed individually via Skype; all other interviews were face to face in a private room while attending hospital or YPAG meeting. Children and young people had a range of long-term health conditions including arthritis, asthma, cancer, as well as neurological and muscular conditions.

### Findings

Based on both rounds of interviews, we describe below what information participants felt was important when considering participation in a trial, their preferences regarding the design, tone and wording of the multimedia websites and how this knowledge informed the development of the websites. A striking overarching finding was that most participants spoke about the need for the views and preferences of children and young people to be central in the design of multimedia websites. As one parent commented:

> It's more important to get across to the teenagers because it's the teenagers who are going through it, you know, the parents are just a by-product of what is happening. (Parent/33)

While the aspiration to prioritise children and young people's views may have shaped participants' accounts, nevertheless, we also outline areas where the various participant groups (children, young people, parents and professionals) differed in their views of the content and design of multimedia websites. Given the gatekeeping role of parents and professionals and potential to constrain children's and young people's access to multimedia websites, taking account of such differences is important.

### Content of the multimedia websites

In the sections that follow, we outline what information participants regarded as important and what they regarded as of little or no importance.

#### That you can leave a trial

A particularly important piece of information for children and young people was knowing that even after joining a trial they could later change their mind and leave. Children and young people who had been in a trial in particular often emphasised the importance of knowing that it was 'okay' to leave the trial at any time. For example, knowing they were free to leave if the treatment subsequently caused significant side effects helped them to feel safe to say 'yes' to a trial. One young person said that this information was her only priority. Professionals also

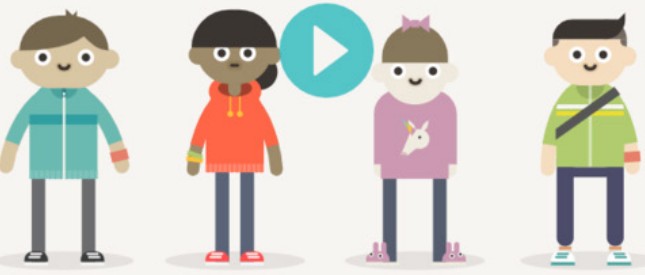

**About the Trial**

The SCIPI trial is testing two ways to give insulin to children and teenagers who have recently found out they have diabetes.

SCIPI is short for Subcutaneous Insulin: Pumps or Injections. In the SCIPI trial we will compare insulin given by injection or a pump. This website provides information about the SCIPI trial to

**Figure 3** Website for young people and their parents

ranked this as an important aspect for families and said that they emphasised it during their initial discussions with families about trials.

> Knowing that you can stop at any time is also good because say you just didn't want it anymore and you were getting bad side effects and you didn't like it, then they could stop it just then and there. (CYP/16)

> If I don't feel comfortable in the trial anymore or if I didn't want to take part because I didn't think it was working for me, could I just stop it – and they were really good at explaining that you could just stop at any time that you wanted and you just have to let them know… so that was one of the only questions that I had really. (CYP/23)

We therefore made the information that participants could leave the trial at any time and do not need to provide a reason prominent within the multimedia websites.

### What is involved if I participate?

Children, young people and their parents wanted to understand what participation in a trial involved. Several who had previously been approached about a trial commented that they would have liked to know more about what was going to be required of them than they

had been provided with when they were approached about a trial. Young people particularly wanted detailed information about the number of clinic or hospital visits, whether they would need to have time away from school and detail about what the treatments involved, including what children would experience during treatment and afterwards. One parent remarked that printed information sheets never focused on details about how participation would affect their child's day-to-day life.

Some children and young people spoke of a pronounced fear of needles and wanted to avoid 'aggressive' or 'scary' portrayals of injections in the multimedia website animations. This fear was not confined to the younger participants: one 15-year-old girl said her main question before deciding to participate in a trial was whether she would receive numbing cream before her injections. Children and young people also wanted information about the taste of oral medication, although professionals noted it can be difficult to give this information in advance for new drugs or formulations. Parents often wanted to know how participating would affect their child emotionally and whether it would give rise to 'undue stress' or 'burden'. Some young people and parents talked about stress as being a crucial consideration but that it was often not covered in the written information they received.

Box 2  Topic guide for round two interviews and focus group discussions

Can we talk about what your first impressions of the multimedia information resource (MMI) are?
Now I would like us to discuss different aspects of the MMI:

**Visual design**
► What do you think about how the MMI looks?
► What parts of the MMI appearance do you like? Which do you not like?

**Structure and navigation**
► How did you find moving through the different parts of the MMI?
► How was it to find and understand information?
► What do you think about the mix of video, text, pictures and animation? Should anything be added or changed?

**Functionality**
► How did the MMI pages load?
► Were there any problems accessing the different parts of the MMI?

**Content**
► What do you think about the information content of the MMI?
► Do you feel it gives you enough information about what is involved in participating in a clinical trial?
► Can you tell me about anything that you think is missing from the MMI?

Is there anything else that you like about the MMI that we have not covered?
Is there anything you particularly liked about the MMI?
Is there anything that you did not like about the MMI?
What should we focus on, if anything, to improve the MMI?
Can you tell me how you think you would use the MMI?

Professionals echoed this, agreeing that most families wanted to know how participating in the trial would impact on them and what was involved.

For young people what's often overlooked is missing out on things like school and social, that's often not mentioned. (CYP/18)

How much time is this going to take that's by far the biggest anxiety that our families have that they're going to have to make extra visits to the hospital or extra tests to do, extra diaries to fill in at home. The burden to them is really important and then safety. (Professional/15)

When developing the multimedia websites, we included detailed information about what trial participation would involve. The websites also make it much easier to convey this detail with visuals and diagrams, helping to clarify what the family could expect if they participate in the trial.

### What is the trial testing?
Children and young people who had first-hand experience of trials often described the science behind their condition and treatment and indicated an interest in knowing how the treatment being trialled was anticipated to work. Parents did not always prioritise such information but some wanted to know whether the treatment had been tested before and, if so, in how many children. Professionals talked about the need to explain the science of the study in simple terms to families in order to convey the rationale for a trial. They emphasised the importance of being clear that professionals do not know the best treatment for a condition and that this is what the trial is trying to find out. In designing the multimedia websites, we therefore made sure that information about what the trial was testing was prominent and provided detailed but accessible information about the trial.

### Risks associated with the trial—and how to inform about these
Most participants talked about the importance of having risks explained but wanted such information to be 'balanced' so as not to unnecessarily concern prospective participants. Parents specifically wanted to know about the possible side effects associated with treatments. However, they also warned that there was sometimes 'far too much information' about risks and that some risks were 'tiny' but were explained in a way that could be 'scary' for young people. Young people and some children wanted to know exactly what they were consenting to in terms of risks and side effects but preferred this to focus on 'likely' risks, rather than being 'overwhelmed' with a list of every 'potential' risk or side effect. Professionals noted that in their experience, families wanted to understand the risks and safety issues associated with being in the trial.

Well risks is mostly for, I'd say mostly for the person taking part because it's going to happen to you if there is any possible risks. Like I know that my drug doesn't always work because I'll have a very, very bad

**Table 2**  Summary of participant characteristics

| Participant group | n | Age, years Mean (range) | Gender | Experience of being approached about a children's trial | Had taken part in a trial |
|---|---|---|---|---|---|
| Children and young people | 21 | 12 (6–19) | 8 male 13 female | 10 | 8 |
| Parents | 24 | – | 8 male 16 female | 13 | 12 |
| Health professionals | 17 | – | 4 male 13 female | n/a | n/a |

flare up and that's kind of a risk. It can't always work but I think it's really important to know that what could happen to you, what you're getting yourself into. That just makes you feel, I don't want to do this or I'm willing to do this. I think that's really important. (CYP/23)

The risk factor thing is a huge thing, you know, to me some of the information sheets I've read I wouldn't do it because I mean potentially some of those side-effects are tiny but yet the way it's put across is so scary that you just wouldn't do it. I know they have to do it, but is there a better way of doing that? (Parent/24)

In the multimedia websites, we therefore provided information about the risks within the key section about 'taking part in the trial'.

### Possible benefits of participating in trials
Parents and young people wanted to know how trial participation might benefit others with the same condition, as well as the possibility of any personal benefit for themselves or their child. One parent emphasised that it was valuable for clinicians to convey this information, because while taking part in a trial is not always 'easy or pleasant', it is 'important' to future patients.

But one thing I do like about trials is that I like to know that the trial is going to benefit somebody in the future. (Parent/36)

That it's just going to help other people if they've got problems. (CYP/16)

Sometimes families believed that their child may get better treatment in a trial, and this gave rise to a concern for some that if their child did not participate in the trial, or left the trial, their quality of care would be reduced. This indicated the continued importance of emphasising that children will receive good care regardless of the decision about trial participation. We therefore made sure that the section of the websites on the potential benefits of the trial was prominent and that it outlined how the trial findings could inform treatments for children in the future.

### Confidentiality
Confidentiality had different meanings for different participants. Some parents expected that their child's data would be shared among those providing care and that there was data sharing among all NHS organisations, including the family's GP, to enable continuity of care. Others felt that it was not important for their GP to be informed of their child's participation in a trial, with one parent saying GPs are 'copied into everything' and implying that such sharing of information happened automatically for care provided in hospital. Young people and parents often said that confidentiality was important, but it was something that they 'assumed' would be observed. Therefore, they felt that confidentiality did not need to be emphasised and that other information was of a higher

priority. Professionals did not feel that information about confidentiality was a priority for families, particularly for children and young people, when deciding to take part.

Well you kind of assume your information is confidential anyway so it's not an issue that you would encounter getting normal health care so you'd assumed the rules wouldn't be different in a research study (CYP/18)… But it's nice to be reassured about it. (CYP/20)

Reflecting these findings, we included information about confidentiality in the multimedia websites, but rather than giving it prominence as its own section on the website, it was included as one of several topics in a question and answer section.

### Who has reviewed or funded the trial
Most children, young people and parents similarly assumed that quality control aspects of trials, such as ethical reviews, and scrutiny of funding sources, were carried out as a matter of course and they trusted the organisations running studies to scrutinise these aspects. However, some young people and parents talked about preferring hospital-led trials over commercially funded trials, implying information on funding sources and who was running a study was of interest to them. Echoing this, some professionals commented that families were more likely to join studies that were being ran by doctors they knew. Therefore, while most did not feel that the above information needed to be prominent in multimedia websites, access to such information could be important for some. Parents also felt it was important to include a relevant logo on the multimedia website, such as an NHS or local hospital logo:

If I saw that NHS logo, for me, it gives it more credibility I think. (Parent/60)

Parents further emphasised the importance of knowing that a website their child was accessing was credible and explained that seeing a logo would reassure them that a website was reputable and safe. Young people generally were not concerned about credibility of the site but liked the inclusion of a logo. Given the range of views, we included information about who had reviewed and funded a study in the multimedia websites but did not make this prominent, and we incorporated logos into the websites.

### Information about payments
Children and young people did not feel that information about the availability of incentive payments would influence their own decisions about participating. However, some worried that providing such payments may lead others to overlook the risks, and therefore that providing any payment to trial participants was 'dodgy', not to mention providing information about such payments. Professionals noted that it was rare for patients to be paid incentives to participate in paediatric trials in the UK. In

contrast, professionals and young people agreed that for studies involving healthy volunteers undertaking multiple blood tests with no personal benefit, providing information about payments would be needed to ensure recruitment. Aside from such studies, most participants did not believe that providing information about payments for participation in a trial was advisable. We positioned information about payments within the 'Questions and Answers' page rather than having its own specific profile on the website.

### Design and tone of the multimedia websites
### Colour
Young people felt that colour in multimedia websites was important to engage the audience but noted that its use needed balance to ensure the website looked 'professional'. Children (aged 6–11 years) and their parents preferred bright colours, whereas young people preferred more muted colours. Some parents and professionals advised avoiding a particular shade of green, which they associated with illness. While participants liked that the animation characters in the prototype websites represented a range of ethnic backgrounds, ensuring the realistic skin tone of characters was a concern for some parents. Reflecting these comments, after the second round of interviews, we changed the skin tone for a few characters in the websites. We also ensured that the colours used in the multimedia website for children was bright (predominantly orange), and the multimedia website for young people was more muted (predominantly teal).

### Layout
Young people talked about the need for the multimedia websites to be well structured and easy to navigate with the main points 'staring right in your face' and having 'simple' headings. They described websites that they liked, and even though these contained a lot of information, it was clearly laid out and easy to read. They liked the relative 'formality' and 'professionalism' of the multimedia websites and wanted the inclusion of characters and other images to balance out the text and make the websites more interesting. We ensured these aspects were included in the multimedia websites.

### Font, characters, quirkiness and details
Young people wanted the font to be easy to read and not in bubble or swirly styles. Indeed, they had little tolerance for fonts that were difficult to read and commented that this would stop them from using a website. Large font size was particularly important for young children, and following their comments, we further increased the font size on the multimedia websites after the second round of interviews.

> Something else that is really important is the type and size of font because it's obvious that people don't think about. If you can't read the font then there is no point to making a website. (CYP/18)

In the first round of interviews, when showing participants the possible character styles for the multimedia websites, participants stated that they wanted the characters to resemble people rather than more abstract 'blobs'. Both children and young people preferred characters that they could 'relate to' and easily recognise the role that a character was representing:

> Maybe you need a nurse that suits a teenager because they [nurse characters in prototype website] seem like they're a children's one with the bear. You need one that you can relate to. (CYP/35)

We revised some of the characters in the websites to ensure that the roles were easily identifiable.

Parents of young children commonly liked characters depicted as very simple shapes, and when elaborating on what it was about these that they liked, it often came down to them being brightly coloured. Parents of young people tended to dislike characters depicted as simple shapes, commenting that these suited children but not young people. However, a few young people in interviews corrected their parents on this, saying that they felt shapes could work, although most preferred characters that looked like people. Overall, children and young people were drawn to characters with brighter clothing and details (eg, red dress with white spots and a T-shirt with a lightning bolt), and we incorporated these details into the final website characters. Quirky or slightly unexpected details often received favourable responses. For example, the draft 'why do we do trials' animation often prompted families of young children to begin talking together about the animation:

> It was funny wasn't it the guy with the pan on his head, it's quite funny. (Parent/41)

> Funny. That was funny with the pan on his head [laugh]. (CYP/43)

In the multimedia websites, we included characters with interesting details and incorporated quirky elements within the other animations to make these engaging for families.

### Animations
Conciseness was important for participants with most saying they would be prepared to watch animations of up to 90s at a sitting, implying that longer animations would not hold their attention. Young people were particularly critical of repetition and some parents also found this distracting. A number of animations were shown to participants in the first round of interviews and focus group discussions. One animation about febrile neutropaenia, depicted using Lego characters and narrated by a child, was popular particularly with children (9–11 years) and their parents. Hearing the child's voice engaged children, even those who did not participate much in the interviews. However, young people often said they felt that this animation style and narration was a little 'young' for them. Another animation, 'Making the right decision

for me', which was narrated by a young person and used sounds to accompany movements and actions, was popular with most children, young people and parents.

> It was so intriguing just like the video and what she was saying all went together really nicely. You could put yourself physically in it and just imagine it. (CYP/20)

However, one group of professionals responded very negatively to this particular animation commenting that it was 'absolutely horrendous' (Professional/3). The animation depicted a young person arguing with her parents when deciding whether to participate, being concerned about not seeing her friends due to research commitments, and the term 'lab rats' was also used at the beginning. These professionals felt the tone was unnecessarily negative and that the content was condescending because it implied that assent was less important than consent. Despite having very different views of this particular animation, parents, professionals, young people and children were united in their overall view that animation was a suitable means of providing information for all participants regardless of their age. Parents often talked about the benefits of seeing and hearing an animation to convey complex information that could otherwise be challenging to explain to a child. For this reason, we developed four generic animations about trials and created a trial specific 'explainer' animation summarising the key features of a particular trial in under 60s.

> I was thinking just then of watching it as an adult that even though it was an animation, you think animations are for children but actually when you look at it, sometimes when you just read something on a page or you've just got somebody speaking at you, it doesn't necessarily go in easily and I would say it doesn't matter that it's an animation like that for an adult watching it even, because you tend to think of it as just for children don't you? But they've made it [Nuffield animation] nice and simple. (Parent/32)

### Narration
Young people were very sensitive to the sound and tone of the narrator's voice. As previously noted, children and their parents liked the example animation narrated by a child:

> I really like it because of the Lego and the child's voice. That catches your attention doesn't it? (Parent/27)

In contrast, young people and their parents felt that a child's voice-over would make it hard for them to relate to the animation and several found it distracting if the pronunciation was imperfect (eg, if the child narrator had difficulty pronouncing some words). We therefore decided to have an adult's voice narrating in the prototype animations. However, on viewing the prototypes, most young people reacted negatively to the narration, commenting that they disliked the 'boring', 'monotonous' or 'robotic' tone of the narrator's voice and that

it stopped them from focusing on the animation itself. Parents did not generally focus on the voice-overs. To address young people's concerns, we asked the narrator to record the narrations again with greater variation in tone.

### Hearing from other families about their experience of trials
When we showed participants a short example video in the round one interviews of a parent and young child talking about participating in a trial investigating light treatment for vitiligo,[27] most parents, young people and children responded favourably, commenting that hearing from those actually going through the trial or treatment was 'reassuring'. Professionals echoed this, commenting that children would want to hear from and see other children who have been in trials, rather than parents or other adults.

> It needs to be people who are actually in trials so then it will put people of my age and younger under reassurance that other people have had a trial before. (CYP/51)

Reflecting these views, we created video clips of trial participants and their parents talking about being in a trial for the multimedia websites.

### Interactivity
Participants differed in their opinions on the value of interactivity in the multimedia websites. They also varied in how they defined interactivity, and this may explain the differences in opinion. For some, interactivity meant including games in multimedia websites, while others felt it referred to being able to ask questions online and receive a response, and others thought it could mean following a pathway through the website. One young patient interpreted websites with multimedia to be interactive because you can 'hear and see things' rather than just read them.

Parents of the youngest children often wanted games, interactive characters and even music and singing included in the multimedia websites and felt that without this their child would not engage. Some professionals agreed that children would like interactivity on multimedia websites, and this could include 'quizzes', 'games' or 'being able to follow a character along a journey on the website'. However, when discussing preferred features of the websites, most young people felt that including interactivity would distract from its purpose unless the interactive features were very relevant. Some parents also felt that interactive aspects such as blogs and forums could be 'negative' and 'unhelpful' unless carefully managed.

> It goes one way or the other I think, either people just ignore it [interactivity] entirely or it's the only thing they look at, neither of which are particularly useful. (CYP/20)

We decided against including interactivity in the multimedia websites, in part due to practical aspects of

developing and maintaining interactive components and due to the varied perspectives about the value of interactivity.

## Wording is important

Careful choice of words was important so as not to scare children and their families. For example, parents, some young people and professionals felt that terms such as 'risk' and 'dangerous' would unnecessarily worry children and young people. To address this, in the multimedia website for younger children, we used a heading of '*Is taking part safe?*' instead of the original '*Is taking part dangerous?*'.

Children and young people also indicated the importance of using terminology familiar to them. For example, the abbreviation FAQs (frequently asked questions, which was used in the draft multimedia websites) meant little to children and young people, who preferred 'Q&A' or 'Questions and Answers'. Young people often spoke positively about the text used in the multimedia websites and liked the simplicity of its wording.

> I like how there's no big fancy medical words [in the website] because I kind of feel like that's kind of really difficult to understand when you're trying to read about what's going to happen and they've got these massive words. You kind of feel like, for people my age and younger, it needs to be easy for us to understand so you actually know what's going to happen to us. (CYP/51)

We revised the wording on the multimedia websites after the second round of interviews to ensure that it was meaningful to children and young people.

## DISCUSSION
### Main findings

We believe this is the first UK study to explore and apply knowledge about the views of stakeholders in developing multimedia websites that aim to help children and young people decide about clinical trial entry. Children, young people and parents prioritised information about what would happen to the participant within the trial, such as how the trial and the treatment being tested would affect them, the time commitment and potential harms and benefits of participation. We incorporated this information in the multimedia website and made the prioritised information prominent. Participants wanted multimedia websites that were easy to use and engaging with plain language and again we incorporated these considerations into the design. They also valued learning from other young patients and their families about what it was like to be in a trial, and we therefore developed and incorporated video clips in the websites with other families talking about their experiences. Participants had many suggestions for how the websites should look, and despite some variation in opinion, they had clear preferences for character styles that resembled people, bright colours for young children

and muted colours for young people, and we have incorporated these preferences in the websites. However, we were not able to incorporate every suggestion requested by participants. For example, we were unable to incorporate interactivity in the multimedia websites, despite this being important to parents of younger children. Furthermore, when the use of animations in the multimedia websites was first suggested, some parents in the round one interviews were concerned that young people might regard these as 'too childish'. However, these concerns were allayed when participants in round two interviews viewed the prototypes, with several commenting that the animations were engaging and provided information in a way that other methods struggled to achieve.

### Our findings in relation to other studies

There is limited previous research about the development of websites (or apps) about healthcare for children and young people.[28–32] Findings from our study are largely in agreement with the information content priorities identified in a previous survey of children and young people in relation to a hypothetical trial,[32] although this survey did not explore priorities for the design of multimedia websites. Findings from previous studies[33–39] of trial participation showed that personal benefit and helping others was important for young people and parents in deciding whether to participate, which also concurs with our findings. However, children, young people and parents in our study also identified information that was of lesser priority to them when deciding about trial participation, including regulatory information and participant payments. Regarding payment, we acknowledge that participants' accounts of what influences them may not necessarily align with influences in practice, although others have also reported that financial reward for trial participation was not a motivation for young people.[38] We did not find privacy to be an important consideration for children and young people, which contrasts a previous survey about hypothetical trial decision making.[32] Our findings on participants' priorities regarding website design, such as font and colour are in line with another study involving young people in the development of a self-management app for asthma[30] and studies showing that adults[40] and young people value clarity in a website providing trial information.[41]

We used interactive techniques such as card sorting to facilitate participants in conveying their information priorities and developed a version of the cards for younger children that used age-appropriate language. Most children engaged with this process and having a parent present seemed to facilitate active participation of children in the interviews. We believe that the use of participatory design in this first phase of the TRECA study has resulted in multimedia websites that are more likely to suit end users in the six future trials. While this remains to be confirmed, previous research shows that resources informed by findings from participatory design are likely to be regarded as relevant and trustworthy by end users.[31 42 43] We have also identified that children, young people, parents and professionals will vary in their views regarding particular aspects

of the multimedia websites. While some might argue that the views of children and young people should take precedence in developing websites, and including some parents in our study, we worked to take account of the perspectives of all parties as parents and professionals may restrict children's and young people's access to websites that they feel are unsuitable.

We did find strong divergences between participants' information priorities and current UK Health Research Authority guidance. Of the topics explored, confidentiality was the area where participants' priorities diverged most from the UK Health Research Authority's guidance, which states that confidentiality should be covered in study information materials. Participants' regarded confidentiality as important but assumed it was observed by researchers as a matter of course and so it was not fundamental to inform their decisions about whether to take part in a trial. We resolved this divergence by including information about confidentiality in the websites but not making it prominent. However, we acknowledge that there may be instances when stakeholders' views cannot be so easily reconciled with guidance (eg, had participants wanted information about risks to be removed or made less prominent). We advise that when such divergences arise, they are dealt with on a case-by-case basis according to the context and reasons for the divergence. The second phase of the TRECA study will involve further development and evaluation of the multimedia websites, including trials of the multimedia websites nested within six UK paediatric clinical trials. Embedded trials, or studies within a trial, enable the evaluation of trial processes such as recruitment techniques.[44]

### Strengths and limitations

The design of our multimedia websites was strengthened by interviewing children, young people, parents and professionals to identify their needs and preferences and use these to inform the development of the multimedia websites. During interviews, we used activities and showed examples of animations, video, websites and character styles to facilitate children's and young people's engagement in the interviews and enable them to say what was important to them. We conducted two rounds of interviews to ensure the multimedia websites could be adapted and refined to incorporate the needs and preferences of users. A potential weakness is that some young children involved in the second round of interviews had difficulty explaining what they liked or disliked about the draft websites, and the final designs may not adequately encompass their preferences. Nevertheless, most children were able to convey their views on the websites. Our sampling was limited to two regions of the UK, and it is unclear how far our findings are transferable to other settings. However, we have tried to describe our methods in sufficient detail so that other researchers can consider the transferability of the findings to their own setting or implement a similar study.

### Future implications

Participants were clear that receiving information via multimedia had several benefits over written information sheets, including enhancing their understanding of what was often complex information.

This is in line with other studies.[11 14 15] The animations seemed to be particularly helpful in this respect as participants found them engaging, and the use of moving visuals and pictures allowed complex ideas to be explained in ways that written text alone cannot achieve. This has implications for how we inform children, young people and their families about trials and addresses the UK Health Research Authority's recommendation to explore non-written methods of providing information about research so that people can access the information most important to them before deciding whether to participate.[13] While this study's focus was developing multimedia websites for paediatric trials, there is potential benefit in using multimedia in other contexts such as explaining healthcare treatment.

### CONCLUSIONS

We drew on findings from interviews with children, young people, parents and professionals to develop the TRECA multimedia websites. The findings showed that children, young people and parents wanted to know about what participation in a trial would involve, what the trial was testing, the potential benefits and risks of taking part and knowing that they could leave the trial if they later changed their minds. Young people and parents felt that regulatory and quality requirements were not priorities for inclusion in websites, although logos of trusted organisations conveyed credibility. It was also important that the websites had the right tone and wording and included a balance of information and pictures, with interesting or quirky details being seen as important to encourage engagement with the websites.

Drawing on a participatory design has helped create multimedia websites that are engaging and provide information in a way that is accessible to children, young people and their families. We will use the insights from this study to develop multimedia websites that include content for six UK trials and evaluate the websites for their impact on recruitment, retention and quality of decision making.

**Author affiliations**
[1]Department of Health Sciences, University of York, York, UK
[2]Department of Health Sciences & Hull York Medical School, University of York, York, UK
[3]NIHR School for Primary Care Research, University of Manchester, Manchester, UK
[4]School of Design, Clothworkers' Central, University of Leeds, Leeds, UK
[5]School of Education, University of Durham, Durham, UK
[6]University of Liverpool, Liverpool, UK
[7]Clinical Research Facility, Alder Hey Children's Hospital, Liverpool, United Kingdom
[8]Patient and Public Involvement member, London, UK
[9]Paediatric Intensive Care Unit, Alder Hey Children's Hospital, Liverpool, UK
[10]Department of Psychological Sciences, University of Liverpool, Liverpool, UK

**Acknowledgements** We gratefully acknowledge all the children, young people, parents and health professionals who participated in the interviews and provided

invaluable details about their needs and preferences of a healthcare trial website. We would like to thank members of our TRials Engagement in Children and Adolescents (TRECA) Study Advisory Group who are not authors on this paper (Professor Carrol Gamble, Professor Michael Beresford and Ms Robyn Challinor), the TRECA Study Steering Committee members (Professor Faith Gibson, Professor Louise Locock, Dr Bob Philips, Dr Louca-Mai Brady and Dr Matt Sydes) and the TRECA Study Patient and Parent Advisory Group for their input into the study. Thank you to those who helped us identify and recruit participants for the interviews. We would like to thank the SCIPI Trial Investigators for allowing us to use content from the SCIPI trial when developing these multimedia websites. The SCIPI trial was funded by the NIHR Health Technology Assessment programme (08/14/39). We would also like to thank the Hi-Light trial investigators for allowing access to their participant video for use in the interviews. We would also like to thank Morph (https://morph.co.uk/), who developed the multimedia websites based on data from each round of interviews.

**Contributors** PK, BY, SH and JMM-K designed the structure of the interviews. JMM-K conducted the interviews and focus group discussions and analysed the data with guidance from BY, KA and PK. JMM-K wrote the manuscript with specific key input from BY, KA and PK. PBo, IW, SH, PBa, CS, JP, RS and DHT all reviewed drafts of the manuscript and provided input to critically revise the manuscript. All authors read and approved the final manuscript.

**Funding** The TRECA study was funded by the NIHR Health Services & Delivery Research Programme (project number 14/21/21).

**Disclaimer** The views expressed are those of the authors and not necessarily those of the NHS, the NIHR or the Department of Health.

**Competing interests** None declared.

**Patient consent for publication** Not required.

**Ethics approval** Yorkshire and the Humber Research Ethics Committee (16/YH/0158), the Health Research Authority (IRAS ID 195396) and the various NHS Trust Research & Development Departments involved approved the study.

**Provenance and peer review** Not commissioned; externally peer reviewed.

**Data sharing statement** No additional data are available.

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
