## [Reviewer comments · BMJ Open]

ARTICLE DETAILS

TITLE (PROVISIONAL)	Supporting children and young people when making decisions about joining clinical trials: qualitative study to inform multimedia website development
AUTHORS	Martin-Kerry, Jacqueline; Knapp, Peter; Atkin, Karl; Bower, Peter; Watt, Ian; Stones, Catherine; Higgins, Steven; Sheridan, Rebecca; Preston, Jenny; Horton Taylor, Danielle; Baines, Paul; Young, Bridget

VERSION 1 – REVIEW

REVIEWER	Els Maeckelberghe University of Groningen, University Medical Center Groningen, The Netherlands
REVIEW RETURNED	21-Jul-2018

GENERAL COMMENTS	This is an important and very timely study. Informing young participants about pediatric trials remains a challenge. This paper presents an exemplary model of how to develop a multimedia website that gives information in a modern and child- and young people friendly way. I have truly enjoyed reading it. The study is described very clearly, and it is "replication-ready". The authors mention that "A potential weakness is that sampling was limited to two regions of the UK, and it is unclear how far our findings are transferable to other settings." I think this so-called potential weakness is overcome due to the detailed and focused method section: the study as such can be replicated both in other regions (in and outside the UK) and other settings (adult trial information). Furthermore, it is a pleasure (and I do not use this word lightly) to see the consistent Public Patient Involvement in this study. What I missed in the paper, was a section about how the assessment of differences in insights was done, e.g. the theme 'confidentiality' was addressed in a Q&A section because participants had different views on it, trusted the professionals etc... In this case, it seems intuitively right to give this information not a prominent place on the website. But, what is behind this intuition? My question actually is of an epistemological nature: whose knowledge prevails and how to decide on that? Imagine both children, young participants and parents would have said it is not very important to give information about what the trial is testing. Or to stretch our imagination even further: risk aren't that important.... Would this information have been referred to a Q&A section? Getting insight in how these decisions are made, is important for further research if you want to get an answer to the question: is this qualitative study to inform multimedia website development transferable?
--

	Continuing this line of thought, another question I want to raise is regarding the use of the website and its implications for changing it or not. Do the authors have any information on how the information is used once it is on the website? From the perspective of the participants: it is one thing to review the multimedia in this study setting for the development of the website; in practice it can be very different: people click and scroll and wander off whilst doing that... Given the technology, it is possible to monitor how long people stay on the website, what items they stay on etc... Again, imagine people only read/watch/access the contact page... And again: based on what type of arguments would adjustments to the site be made? Is it because there is an understanding about what kind of information people ought to have seen or does it suffice to have the information available and people, even children and young people, are allowed to make decisions without much thought, as long as we can ensure that they had easy access to good and tailor made information? But this topic is probably what the authors are going to investigate in the next study given their intention to "evaluate the websites for their impact on recruitment, retention and quality of decision-making". I am looking forward reading those results.
--	--

REVIEWER	Lorraine Smith University of Sydney,Australia
REVIEW RETURNED	16-Aug-2018

GENERAL COMMENTS	This is a very welcome addition to the body of research on participating in clinical trials, particularly as it involves children. My comments are as follows:  1. Intro, last para, page 4, line 20: can you explain your interpretation of the word 'constrain' so that the reader has some idea of what ways parents and professionals might 'constrain' children's access to information? 2. Methods, first sentence, page 4, line 28: typo 3. Data collection, page 5, line 17: what sources of information did you use to inform the topic-guided, semi-structured interviews? 4. First round of interviews, page 5, lines 37-40: the exercise you used to help facilitate discussion amongst participants would have been challenging particularly for the younger participants, even those aged 9 and older. Can you comment on this in the discussion? 5. Box 2, page 7, line 3: how easy would this have been for 6-8 year olds? As per point #4, can you comment on this? Perhaps add something to the limitations section in the Discussion. 6. Box 2, page 7, line 5: word missing - 'what' 7. Results, Participants, first para, page 8, lines 30-50: this might look better as a flow chart. As it currently reads it's quite complicated. 8. Table 2, pages 9 and 10: these data could be presented more succinctly by aggregating them. 9. Possible benefits of participating in trials, page 14, line 28: typo 10. Animations, page 18, line 17: it is stated that animations were popular with young children - but it is stated earlier that young children did not participate in the first round of interviews (see Second round of interviews, page 6, lines 46/47. 11. Hearing from other families ..., page 19, lines 36-40: again I'm confused by the reference to round one interviews and the inclusion of young children which I thought excluded the young ones.
--

REVIEWER	Jamie Roberts Duke University School of Medicine, USA
REVIEW RETURNED	20-Aug-2018

GENERAL COMMENTS	This is a well-written paper with very interesting findings and I have very few comments -- all of them editorial in nature (see attached). Thank you for the opportunity to review this paper -- I look forward to seeing it published. - The reviewer provided a marked copy with additional comments. Please contact the publisher for full details.
---

VERSION 1 – AUTHOR RESPONSE

Reviewer: 1

Reviewer Name: Els Maeckelberghe

Institution and Country: University of Groningen, University Medical Center Groningen, The Netherlands

Please state any competing interests or state 'None declared': None declared

Please leave your comments for the authors below

This is an important and very timely study. Informing young participants about pediatric trials remains a challenge. This paper presents an exemplary model of how to develop a multimedia website that gives information in a modern and child- and young people friendly way. I have truly enjoyed reading it.

Thank you for this feedback.

The study is described very clearly, and it is "replication-ready". The authors mention that "A potential weakness is that sampling was limited to two regions of the UK, and it is unclear how far our findings are transferable to other settings." I think this so-called potential weakness is overcome due to the detailed and focused method section: the study as such can be replicated both in other regions (in and outside the UK) and other settings (adult trial information).

Thank you. We have added a sentence to reflect that the methods are sufficiently detailed to enable others to consider the transferability of the findings to their setting or locations, or to replicate this work.

Furthermore, it is a pleasure (and I do not use this word lightly) to see the consistent Public Patient Involvement in this study.

Thank you for this comment. Our PPI members have been actively engaged with the study and provided important input throughout.

What I missed in the paper, was a section about how the assessment of differences in insights was done, e.g. the theme 'confidentiality' was addressed in a Q&A section because participants had different views on it, trusted the professionals etc... In this case, it seems intuitively right to give this

information not a prominent place on the website. But, what is behind this intuition? My question actually is of an epistemological nature: whose knowledge prevails and how to decide on that? Imagine both children, young participants and parents would have said it is not very important to give information about what the trial is testing. Or to stretch our imagination even further: risk aren't that important.... Would this information have been referred to a Q&A section? Getting insight in how these decisions are made, is important for further research if you want to get an answer to the question: is this qualitative study to inform multimedia website development transferable?

We have added some content within the discussion to acknowledge this issue(pages 21-22).While we did not have any irreconcilable differences in this study between participants' views on what information to prioritise in the websites and UK ethics guidance, we note that such divergences may arise. However, it is hard to make generalised recommendations beyond our study findings and we advise that such divergences be considered on a case by case basis, according to the issue on which there is divergence and the reasons for the divergence.

Continuing this line of thought, another question I want to raise is regarding the use of the website and its implications for changing it or not. Do the authors have any information on how the information is used once it is on the website? From the perspective of the participants: it is one thing to review the multimedia in this study setting for the development of the website; in practice it can be very different: people click and scroll and wander off whilst doing that... Given the technology, it is possible to monitor how long people stay on the website, what items they stay on etc... Again, imagine people only read/watch/access the contact page...

This will be a focus of Phase two of the TRECA study and we will use Google Analytics to examine how the websites are accessed (which pages are viewed, how long people spend on each page, video, animation etc). The findings will be reported in future planned publications about the websites used in Phase two.

And again: based on what type of arguments would adjustments to the site be made? Is it because there is an understanding about what kind of information people ought to have seen or does it suffice to have the information available and people, even children and young people, are allowed to make decisions without much thought, as long as we can ensure that they had easy access to good and tailor made information? But this topic is probably what the authors are going to investigate in the next study given their intention to "evaluate the websites for their impact on recruitment, retention and quality of decision-making".

I am looking forward reading those results.

Thank you again for your interest in the study. We anticipate that the design of the multimedia websites enable participants to access the websites in a way that suits them and find information according to their needs and preferences. One aspect we are particularly interested in for Phase 2 of TRECA is whether the websites change the quality of decision-making by enabling potential participants to find and access information in a way that suits their own preferences and needs. However, for all trials in Phase 2 of TRECA, families will have discussions with health professionals before deciding to enter the trials and for many, such discussions will be the main source of information for decision-making [1]. The websites, while important, are supplementary to the discussions and we therefore feel it is appropriate for families to engage with the materials in a way that best meets their needs.

[1] Shilling V, Williamson P R, Hickey H, et al. Processes in recruitment to randomised controlled trials of medicines for children (RECRUIT): a qualitative study. *Health Technol Assess* 2011;15:1-116.

Reviewer: 2

Reviewer Name: Lorraine Smith

Institution and Country: University of Sydney, Australia

Please state any competing interests or state 'None declared': None declared

Please leave your comments for the authors below

This is a very welcome addition to the body of research on participating in clinical trials, particularly as it involves children.

Thank you for this feedback.

My comments are as follows:

1. Intro, last para, page 4, line 20: can you explain your interpretation of the word 'constrain' so that the reader has some idea of what ways parents and professionals might 'constrain' children's access to information?

This statement was referring to the role of parents and professionals as a gatekeeper to children accessing information. We have amended the text on page 4 and added a reference to clarify this point.

2. Methods, first sentence, page 4, line 28: typo

This has been corrected.

3. Data collection, page 5, line 17: what sources of information did you use to inform the topic-guided, semi-structured interviews?

We used a systematic review about what potential (adult) participants want to know about research (Kirkby et al 2012) and the Webby criteria for websites for the first round of interviews. This detail is included in the sections about 'First round of interviews', on page 5, and page 6, respectively.

For round two topic guides, we developed these based on with input from qualitative methodologist (BY) with experience of researching families' decisions about clinical trial participation and education expert (SH). We have added this information into the manuscript on page 6.

4. First round of interviews, page 5, lines 37-40: the exercise you used to help facilitate discussion amongst participants would have been challenging particularly for the younger participants, even those aged 9 and older. Can you comment on this in the discussion?

Almost all children actually engaged with this process. We used age-appropriate wording on the cards for younger children. In joint interviews, the parent would sometimes talk to the child about the cards and this helped the children. Most children seemed to enjoy the activity – even those at the youngest end of the age spectrum – and some talked through their decisions as they placed the cards. We have added some additional comment on this in the discussion on page 21.

5. Box 2, page 7, line 3: how easy would this have been for 6-8 year olds? As per point #4, can you comment on this? Perhaps add something to the limitations section in the Discussion.

Almost all children were able to comment on the websites. However there were a few children who struggled in round two to comment on aspects of the draft websites or animations. Some children had difficulty identifying aspects they disliked but all except two young children were able to point out aspects they liked. Having the interviews in interactive format with concrete materials to talk about helped children convey their views about the websites. We have however noted this as a potential limitation on page 22.

6. Box 2, page 7, line 5: word missing - 'what'

This has been added.

7. Results, Participants, first para, page 8, lines 30-50: this might look better as a flow chart. As it currently reads it's quite complicated.

It was difficult to create a flow chart as there were a number of entry points in which people participated (i.e. in round 1 or round 2) and we felt that a flow chart may also be difficult to explain the participants. We have however re-written the paragraph (on page 8) to explain the process more clearly.

8. Table 2, pages 9 and 10: these data could be presented more succinctly by aggregating them.

We have provided a summary table as table 1 (replaces previous table 2) and included the detailed table as an appendix for those who want to see characteristics at the level of individual participants.

9. Possible benefits of participating in trials, page 14, line 28: typo

This has been corrected.

10. Animations, page 18, line 17: it is stated that animations were popular with young children - but it is stated earlier that young children did not participate in the first round of interviews (see Second round of interviews, page 6, lines 46/47).

Here we are referring to children aged 9-11 years who participated in the first rounds of interviews and saw the example animations (described in methods on page 5-6); as well as referring to younger children aged 6-8 years who viewed the draft TRECA animations (described on page 7). We have revised the wording in the manuscript to make this clearer and now use 'young children' only for 6-8 year olds, the term 'children' for those aged 9-11 years, and young people are 12 years and above.

11. Hearing from other families ..., page 19, lines 36-40: again I'm confused by the reference to round one interviews and the inclusion of young children which I thought excluded the young ones.

This sentence was referring to children aged 9-11 years as they reviewed animations in round 1 (page 5-6 of methods). As above, we recognise that the terminology was confusing and we have made changes within the manuscript to standardise the terms used.

Reviewer: 3

Reviewer Name: Jamie Roberts

Institution and Country: Duke University School of Medicine, USA

Please state any competing interests or state 'None declared': None

Please leave your comments for the authors below

This is a well-written paper with very interesting findings and I have very few comments -- all of them editorial in nature (see attached). Thank you for the opportunity to review this paper -- I look forward to seeing it published.

Thank you for your feedback. We have addressed these issues within the manuscript.

We have also made minor edits which did not change the meaning of the content, using tracked changes, to improve the clarity and expression.

VERSION 2 – REVIEW

REVIEWER	Els Maeckelberghe University of Groningen, University Medical Center Groningen, The Netherlands
REVIEW RETURNED	04-Oct-2018

GENERAL COMMENTS	All issues addressed in the review are adequately answered.
---

REVIEWER	Lorraine Smith University of Sydney
REVIEW RETURNED	04-Oct-2018

GENERAL COMMENTS	All of my comments have been addressed, thank you. A great paper which deserves publication.
--

REVIEWER	Jamie Roberts Duke Clinical Translational Science Institute
REVIEW RETURNED	05-Oct-2018

GENERAL COMMENTS	This is a well-written manuscript and an important contribution to the field, detailing how participatory design and qualitative research can inform how we inform and educate young people about research opportunities.
---